# Association between the 2018 WCRF/AICR and the Low-Risk Lifestyle Scores with Colorectal Cancer Risk in the Predimed Study

**DOI:** 10.3390/jcm9041215

**Published:** 2020-04-23

**Authors:** Laura Barrubés, Nancy Babio, Pablo Hernández-Alonso, Estefania Toledo, Judith B. Ramírez Sabio, Ramón Estruch, Emilio Ros, Montserrat Fitó, Angel M Alonso-Gómez, Miquel Fiol, Jose Lapetra, Lluís Serra-Majem, Xavier Pintó, Miguel Ruiz-Canela, Dolores Corella, Olga Castañer, Manuel Macías-González, Jordi Salas-Salvadó

**Affiliations:** 1Universitat Rovira i Virgili, Departament de Bioquímica i Biotecnologia, Unitat de Nutrició Humana, Hospital Universitari San Joan de Reus, Institut d’Investigació Pere Virgili (IISPV), 43201 Reus, Spain; laura.barrubes@urv.cat (L.B.); jordi.salas@urv.cat (J.S.-S.); 2Consorcio CIBER, M.P. Fisiopatología de la Obesidad y Nutrición (CIBERObn), Instituto de Salud Carlos III (ISCIII), 28029 Madrid, Spain; etoledo@unav.es (E.T.); restruch@clinic.cat (R.E.); eros@clinic.cat (E.R.); mfito@imim.es (M.F.); angelmaria.alonsogomez@osakidetza.eus (A.M.A.-G.); miguel.fiol@ssib.es (M.F.); joselapetra543@gmail.com (J.L.); lluis.serra@ulpgc.es (L.S.-M.); xpinto@bellvitgehospital.cat (X.P.); mcanela@unav.es (M.R.-C.); dolores.corella@uv.es (D.C.); ocastaner@imim.es (O.C.); mmacias.manuel@gmail.com (M.M.-G.); 3Department of Endocrinology and Nutrition, Virgen de la Victoria University Hospital, University of Malaga (IBIMA), 29071 Málaga, Spain; 4Department of Preventive Medicine and Public Health, University of Navarra-Navarra Institute for Health Research, 31009 Pamplona, Spain; 5Oncology Service, Sagunto Hospital, 46520 Valencia, Spain; jbramire@uv.es; 6Department of Preventive Medicine, University of Valencia, 46010 Valencia, Spain; 7Department of Internal Medicine, Hospital Clínic, University of Barcelona, 08036 Barcelona, Spain; 8August Pi i Sunyer Biomedical Research Institute (IDIBAPS), 08036 Barcelona, Spain; 9Lipid Clinic, Endocrinology & Nutrition Service, Hospital Clínic, University of Barcelona, 08036 Barcelona, Spain; 10Cardiovascular Risk and Nutrition Research (REGICOR Group), Institut Hospital del Mar d’Investigacions Mèdiques (IMIM), 08003 Barcelona, Spain; 11Bioaraba Health Research Institute, Osakidetza Basque Health Service, Araba University Hospital, University of the Basque Country UPV/EHU, 01007 Vitoria-Gasteiz, Spain; 12Balearic Islands Health Research Institute (IdISBa) and Son Espases Hospital, 07120 Palma, Spain; 13Department of Family Medicine, Research Unit, Distrito Sanitario Atención Primaria Sevilla, Centro de Salud Universitario San Pablo, 41013 Sevilla, Spain; 14Research Institute of Biomedical and Health Sciences, University of Las Palmas de Gran Canaria, 35017 Las Palmas, Spain; 15Lipids and Vascular Risk Unit, Internal Medicine, Hospital Universitario de Bellvitge, 08907 Hospitalet de Llobregat, Spain

**Keywords:** WCRF/AICR score, low-risk lifestyle index, colorectal cancer, PREDIMED, lifestyle patterns

## Abstract

Limited longitudinal studies have been conducted to evaluate colorectal cancer (CRC) incidence based on the updated 2018 World Cancer Research Fund/American Institute for Cancer Research (WCRF/AICR) recommendations or other global lifestyle indices, and none in aged populations at high cardiovascular risk. We aimed to assess the association between CRC incidence and adherence to two emerging lifestyles indices (2018 WCRF/AICR score and another low-risk lifestyle (LRL) score comprising smoking status, alcohol consumption, physical activity, diet, and body mass index) in the Spanish PREvencion con DIeta MEDiterranea (PREDIMED) cohort. We studied 7216 elderly men and women at high cardiovascular risk. The 2018 WCRF/AICR and LRL scores were calculated. Multivariable Cox proportional regression models were fitted to estimate the HRs (hazard ratios) and 95% confidence intervals (CIs) for incident CRC events. During a median interquartile range (IQR) follow-up of 6.0 (4.4–7.3) years, 97 CRC events were considered. A significant linear association was observed between each 1-point increment in the WCRF/AICR score (score range from 0 to 7) and CRC risk (HR (95% CI) = 0.79 (0.63–0.99)). Similarly, each 1-point increment in the LRL score (score range from 0 to 5) was associated with a 22% reduction in CRC risk (0.78 (0.64–0.96)). Adhering to emergent lifestyle scores might substantially reduce CRC incidence in elderly individuals. Further longitudinal studies, which take different lifestyle indexes into account, are warranted in the future.

## 1. Introduction

Worldwide, colorectal cancer (CRC) is an important public health problem since it is the second most commonly occurring cancer in women and the third in men. CRC is also the second most common cause of cancer death in both sexes globally. In 2018, there were over 1.8 million new CRC cases, and the global burden is expected to increase further due to the growth of the aging population and the adoption of westernized behaviors and lifestyles [1].

There is convincing evidence that some dietary components (i.e., processed meat and alcohol intake) and body fatness are modifiable risk factors, contributing to the development of CRC, whereas physical activity decreases the risk [2].

Since foods are not consumed in isolation and have additive or synergistic health-related effects, the current literature focuses on examining diet as a multidimensional exposure [3]. Several a priori defined food groups and general index-based dietary patterns have been associated with lower CRC risk, supporting the hypothesis that high overall diet quality is associated with decreased CRC risk [4,5].

Moreover, since both diet quality and other environmental factors have been shown to play an important role in the development of chronic diseases, healthy lifestyle indices have emerged. Previous work on CRC nutritional epidemiology has mainly focused on the World Cancer Research Fund/American Institute for Cancer Research (WCRF/AICR) score [6]. Nonetheless, very little is known about other lifestyle indices and their association with the risk of this type of cancer [7].

Some prospective cohort studies [6,8,9,10,11,12,13] and case-control studies [14,15,16] have evaluated the associations between CRC risk and adherence to cancer-specific nutritional recommendations with inconsistent results. However, limited longitudinal studies [13,16] have been conducted to assess associations based on the updated 2018 WCRF/AICR recommendations.

Another recent score by Li and coworkers [7], in the context of the Nurses’ Health Study and the Health Professionals Follow-up Study, composed of five modifiable lifestyle factors (smoking status, alcohol consumption, physical activity, diet, and body mass index (BMI)), has consistently shown significant inverse associations with all-cause mortality, including cancer and cardiovascular disease (CVD) mortality and also with CVD incidence. Recently, this index has been also related to an increased healthy life expectancy (i.e., free of chronic diseases) [17]. However, information regarding this score with cancer incidence is lacking [7,18].

Lifestyle scores-generated a priori, according to current scientific knowledge, allows us to examine the potential combined effect of the individual score components on the incidence of different diseases. To the best of our knowledge, no previous study has evaluated the association between the aforementioned lifestyle score (low-risk lifestyle (LRL) score from now on) and the risk of developing CRC.

In an attempt to know whether using scores based on overall lifestyle patterns may be useful to prevent CRC in aging individuals, we aimed to evaluate the associations between adherence to the 2018 WCRF/AICR and LRL scores with CRC incidence in elderly Spanish individuals at high CVD risk. Our secondary objective was to evaluate the associated CRC risk for every individual component of each score.

## 2. Materials and Methods

### 2.1. Study Design

We conducted a prospective, longitudinal, observational cohort study within the frame of the PREDIMED (PREvencion con DIeta MEDiterranea) study. Briefly, the PREDIMED study is a multicenter, parallel-group controlled trial designed to assess the effect of a traditional Mediterranean Diet (MedDiet) on the primary prevention of CVD [19]. The PREDIMED Project was approved by the “Comité Ético de lnvestigación Clínica Hospital Clínic” from Barcelona (Project identification register: 2002–1244; date of approval: 16 July 2002).

The design and results of the PREDIMED trial with respect to the primary endpoint have been reported elsewhere [20]. Before starting the study, all participants provided informed consent. The Institutional Review Boards of each recruitment center approved the protocol. Even though for the main outcome of CVD, the trial was completed after a median follow-up of 4.8 years, we analyzed data based on the extended follow-up until December 2012.

### 2.2. Participants

A total of 7447 individuals from primary care centers were recruited to the PREDIMED trial between 2003 and 2009. Eligible participants were community-dwelling men (aged 55–80 years) and women (aged 60–80 years) free from CVD at baseline but who were at high risk because they had either type 2 diabetes (T2D) or at least three of the following cardiovascular risk factors: current smoking, hypertension, hypercholesterolemia, low high-density lipoprotein cholesterol, overweight/obesity, or family history of premature coronary heart disease.

Exclusion criteria were the presence of any severe chronic illness, malignant tumors diagnosis in the last five years prior to the recruitment, alcohol or drug abuse, a BMI ≥ 40 kg/m^2^, and allergy or intolerance to olive oil or nuts. Participants were allocated to one of the three intervention groups: Mediterranean Diet (MedDiet) supplemented with nuts, MedDiet supplemented with extra virgin olive oil, or advice to reduce all sources of fat (control group). Energy restriction and physical activity were not encouraged in any group during the intervention. For this analysis, those participants who had implausible baseline daily energy intake values (<500 or >3500 kcal/day for women or <800 or >4000 kcal/day for men) and those who did not complete the baseline food frequency questionnaire (FFQ) were excluded.

### 2.3. Ascertainment of Incident and Fatal CRC

New CRC events were defined as the first invasive CRC according to the International Classification of Diseases for Oncology topographical codes C18.0–C20.9. The results of the histological examination were considered confirmatory in most events (*n* = 67). Events were identified from the following sources: a review of all the medical records by a panel of physicians and researchers blinded to the intervention, at both primary healthcare and hospital level, and the national death index. The Endpoint Adjudication Committee, whose members were also blinded to the intervention, determined the cause of death, confirmed major events, and updated the endpoints of the PREDIMED study on a yearly basis.

### 2.4. Dietary Assessment

A validated semi-quantitative FFQ [21], which included 137 food items, was used to assess the dietary habits of participants in face-to-face interviews conducted by trained dietitians. The frequency of consumption of food items was asked on an incremental scale with 9 levels (never or almost never; 1–3 servings/month; 1, 2–4, and 5–6 servings/weeks; and 1, 2–3, 4–6, and >6 servings/day). We used Spanish food composition tables to estimate energy and nutrient intake [22,23].

### 2.5. Other Lifestyle Variables Assessment

Trained personnel took anthropometric measurements (weight, height, and waist circumference). To measure weight and height, calibrated scales and a wall-mounted stadiometer were used, respectively, with participants wearing light clothing and no shoes. Waist circumference was measured midway between the lowest rib and the iliac crest using an anthropometric tape. All anthropometric variables were measured annually. BMI was calculated by dividing the weight (kg) by the square of the height (m^2^). Blood pressure was measured with a validated oscillometer (Omron HEM705CP, Hoofddorp, The Netherlands) in triplicate with a 5 min interval between each measurement.

The validated Spanish version of the Minnesota leisure-time physical activity (LTPA) questionnaire [24,25] was used to assess the amount and intensity of LTPA. The questionnaire consisted of 67 activities divided into 9 sections. The participants were asked to complete the form, reporting the number of days and minutes/day they had performed the activities during the previous week and year. Physical activity was quantified in the metabolic equivalent of tasks per minute per day (METs min/day). This unit was calculated by multiplying the METs assigned to each activity and their mean duration in minutes per day. LTPA was classified as light (intensity <4 METs), moderate (intensity 4–5.5 METs), and vigorous (intensity ≥6 METs). This questionnaire was completed during a baseline visit and annually thereafter. Moderate and vigorous intensity levels were combined into one category for purposes of analysis. A general questionnaire about lifestyle variables, such as smoking status or education level, medical history, and medication use, was completed and recorded at baseline and yearly thereafter.

### 2.6. 2018 WCRF/AICR Score Operationalization

We constructed a 7-point score based on the 2018 WCRF/AICR recommendations for cancer prevention [2]. The score components were (1) healthy weight, (2) physical activity, (3) plant foods, (4) fast food and processed foods, (5) red and processed meat, (6) sugar-sweetened beverages, and (7) alcohol. Detailed information on the score construction is shown in Appendix A. The cut-off points for each component were based on 2018 WCRF/AICR recommendations when available or previously published literature otherwise. For each of the recommendations, we assigned 1 point when the recommendation was met, 0.5 points when it was partially met, and 0 points when it was not met. For those components with sub-recommendations, the considered component score was the average of the sub-recommendation scores. The mean score for each component was between 0.36 (red and processed meat consumption) and 0.70 (sugar-sweetened drink intake) points. The final index was the sum of all the components and ranged from 0 to 7. Higher scores indicated better adherence to cancer prevention recommendations.

### 2.7. Low Risk Lifestyle Score Operationalization

The LRL score is a 5-component index that was developed to assess the impact of healthy lifestyle factors on any-cause, cardiovascular, and cancer mortality in the US population [7]. We constructed this score in terms of adherence to the following LRL-related factors: (1) never smoking, (2) healthy weight, (3) regular physical activity, (4) healthy diet, and (5) moderate alcohol consumption. Detailed information on the score operationalization is presented in Appendix A. For each risk factor, 1 point was given if the participant met the criterion for low risk or 0 points otherwise. The mean score for each component was between 0.07 (healthy body weight) and 0.62 (never smoking) points. The final score was the sum of all components (score range from 0 to 5), with higher scores indicating a healthier lifestyle.

### 2.8. Statistical Analyses

For each participant, we calculated the follow-up time as the interval between the date of randomization and the date of CRC diagnosis, death from any cause, or the date of the last contact visit, whichever came first. The baseline characteristics of the participants were expressed as medians and (IQR) for continuous variables, and percentage (%) and number (*n*) for categorical variables. Chi-square and *t*-Student tests were used to assess differences in the baseline characteristics between CRC incident events and non-events. Continuous variables were normally distributed.

Multivariable time-dependent Cox proportional regression models were used to evaluate the associations between 2018 WCRF/AICR and LRL scores at baseline and the risk of developing CRC. Results were the hazard ratios (HRs) and their 95% confidence intervals (CIs) for the comparison between the highest vs. the lowest quantile for each overall score (quartile (Q) 4 vs. Q1 for 2018 WCRF/AICR score, and tertile (T) 3 vs. T1 for LRL score). Quantiles for each index were calculated considering the distribution of the variable in the analyzed population. The highest vs. the lowest categories for each individual component and their association with CRC risk were also compared. We used robust estimates of the variance to correct for potential intra-cluster correlation.

To calculate the associated CRC risk with both lifestyle scores, three different Cox regression models were used. The crude model was a univariate model. Model 1 was adjusted for age (years, as a continuous variable) and sex. Model 2 comprised model 1 additionally adjusted for the intervention group, family history of cancer (yes/no), education level (primary or secondary/high school university or graduate), history of diabetes (yes/no), baseline energy intake (Kcal/day, as a continuous variable), and treatment with aspirin (yes/no) at baseline. Model 2 for the 2018 WCRF/AICR score was further adjusted for smoking (current (yes/no), former (yes/no), or never (yes/no) smoker). The CRC risk associated with categories of the individual components of the 2018 WCRF/AICR score at baseline was additionally and mutually adjusted (model 3) for the other individual components at baseline: healthy weight (0, 0.25, 0.5, >0.75 point), physical activity (0, 0.5, 1 point), consumption of plant foods (0–0.5, >0.5–0.75, >0.75 point), fast food and processed foods (0, 0.5, 1 point), red and processed meat (0, 0.5, 1 point), sugar-sweetened beverages (0, 0.5, 1 point), and alcohol intake (0, 0.5, 1 point). Model 3 for the LRL components at baseline was model 2 plus the other individual LRL components at baseline: BMI ≥18.5 and <25 kg/m^2^ (yes/no), never smoker (yes/no), low-risk alcohol consumption (yes/no), alternate healthy eating index (AHEI)-2010 score ≥P60 (yes/no), and moderate to vigorous physical activity (MVPA) ≥150 minutes/wk (yes/no). All models were stratified by recruitment center. The association with CRC for each 1-point increment in the WCRF/AICR and the LRL scores were also calculated by adding the score variable as continuous in the Cox regression models.

Statistical interactions between quantiles of both WCRF/AICR and LRL scores and potential confounders, such as age (years, as continuous), sex, and diabetes status, were evaluated by including interaction terms in the full-adjusted models. There were no statistical interactions between the WCRF/AICR and LRL scores, age, sex, and diabetes status. Linear trend tests were conducted by assigning the median value of each quantile of scores and individual score components and then using it as a continuous variable. In secondary analyses, we explored the contribution of certain relevant variables to all primary analyses of the scores. Thus, we conducted subgroup analyses by age (<67 or ≥67 y old), sex (men or women), and T2D status (prevalent or non-prevalent).

To test the robustness of our findings, we used a mixed-effects Cox regression model to take into account repeated measures of the covariates and the participants’ deviances into the model. We used a robust estimate of the variance to account for intra-cluster correlation within the same families or clinics (as previously reported in [20]) and the repeated measures of the covariates as the fixed effects. However, the variable for the different scores was set at baseline for comparison purposes with our main analysis. All *p*-values were two-sided, and a *p*-value < 0.05 was considered statistically significant. Analyses were performed using the STATA (14.0, StataCorp LP, Lakeway Drive, TX. USA) and R (v. 3.5.1) software (GNU General Public License, Boston, MA, USA).

## 3. Results

During a median (IQR) follow-up of 6.0 (4.4–7.3) years, we documented 101 CRC events. Of the 7447 participants, we excluded those with energy intake values outside the pre-specified limits (*n* = 153) and those lacking baseline FFQ (*n* = 78). Finally, 7216 individuals and 97 new CRC events were included in our analysis.

### 3.1. Baseline Characteristics of the Participants

The general baseline characteristics of the PREDIMED population are shown in Table 1. The median age of the participants was 67 years, 57% of whom were women, and 49% of whom had prevalent T2D. The median (IQR) age at cancer diagnosis was 72.4 (66.8–76.4) years. The median (IQR) 2018 WCRF/AICR score in the whole population was 3.8 (3.3–4.4) points, with no statistically significant differences between events/non-events (*p*-value = 0.110). The overall median (IQR) for the LRL score was 2 (1–2) points, with CRC events showing statistically significant lower scores than non-events (*p*-value = 0.021). Among the participants who developed CRC, there were significantly more men than women compared with non-events (58.8% vs. 41.2%, respectively; *p*-value = 0.001). In addition, those who did not develop CRC were more likely to be never-smokers in comparison to new CRC cases (61.7% vs. 47.4%, respectively; *p*-value = 0.011).

### 3.2. World Cancer Research Fund/American Institute for Cancer Research and Low Risk Lifestyle Scores and Risk of Colorectal Cancer

Table 2 shows the associations (HRs and 95% CIs) for CRC incidence with the overall 2018 WCRF/AICR and LRL scores at baseline. Statistically significant linear associations were observed between a 1-point increment in the WCRF/AICR score and CRC risk in all statistical models (HR _model 2_ (95% CI) = 0.79 (0.63–0.99); *p* for trend = 0.045). Similarly, each 1-point increment in the LRL score was associated with a 22% lower CRC risk (HR _model 2_ (95% CI) = 0.78 (0.64–0.96); *p* for trend = 0.017).

Along the same lines, when we analyzed the associations between CRC incidence and categories of adherence to each overall score (Figure 1), high adherence to WCRF/AICR recommendations was inversely associated with CRC risk (HR Q4 vs. Q1 (95% CI) = 0.52 (0.27–0.99); *p* for trend = 0.130) compared to the reference category. For adherence to the LRL score, the reduction in CRC risk remained statistically significant (HR T3 vs. T1 (95% CI) = 0.48 (0.26–0.86); *p* for trend = 0.007) when analyzed as categorical. A sensitivity analysis using mixed-effects Cox regression models mirrored our results for the WCRF/AICR (HR Q4 vs. Q1 (95% CI) = 0.57 (0.32–0.82)) and LRL scores (HR T3 vs. T1 (95% CI) = 0.44 (0.20–0.68)). Baseline characteristics of the participants and the number of CRC cases according to quantiles of each score are shown in the Appendix A.

For each 1-point increment in the number of components of the LRL score, there was a statistically significant 23% lower risk of CRC (HR _model 2_ (95% CI) = 0.77 (0.62–0.95); *p* for trend = 0.016) (Appendix A).

### 3.3. Individual Components of the Scores and Risk of CRC

Table 3 shows the mutually adjusted HRs and 95% CIs for CRC risk associated with categories of individual components of the WCRF/AICR score at baseline. In model 3, those individuals with the maximum score (1 point) for the sugar-sweetened beverages component exhibited a significantly lower risk of developing CRC (HR (95% CI) = 0.42 (0.19–0.93); *p* for trend = 0.192) than those with the lowest score (0 points). Non-statistically significant associations with CRC risk were observed for the other score components (healthy body weight, physical activity and consumption of plant foods, fast food and processed foods, red and processed meat, and alcohol intake).

Table 4 shows the HRs and 95% CIs for CRC risk associated with different categories of individual LRL components at baseline. In the crude model, participants who had never smoked presented a statistically significant linear decreased risk of developing CRC (HR (95% CI) = 0.55 (0.36–0.83); *p* for trend = 0.004) in comparison to current and former smokers. Nonetheless, when the model was further adjusted for confounding variables, the association was attenuated and became non-significant (HR (95% CI) = 0.67 (0.40–1.12); *p* for trend = 0.128). Non-significant inverse associations between the other LRL recommendations (BMI, alcohol intake, physical activity, and diet) and CRC risk were observed. 

We also observed statistically significant differences across age, sex, and prevalent T2D status (Appendix A). The inverse association between CRC risk and the WCRF/AICR score (HR _model 2_ (95% CI) = 0.71 (0.50–0.99; *p* for trend = 0.050) and the LRL score (HR _model 2_ (95% CI) = 0.76 (0.58–0.99; *p* for trend = 0.048) was stronger in those participants above the median age (≥67 years) than in those below. Associations for the LRL score were stronger in women (HR _model 2_ (95% CI) = 0.69 (0.50–0.95; *p* for trend = 0.023) than in men (HR _model 2_ (95% CI) = 0.84 (0.65–1.08; *p* for trend = 0.174). However, for the 2018 WCRF/AICR score, there were no significant differences by sex. Those participants presenting T2D at baseline exhibited a stronger association between the WCRF/AICR score and CRC risk (HR _model 2_ (95% CI) = 0.71 (0.53–0.96; *p* for trend = 0.024) in comparison to non-prevalent T2D individuals (HR _model 2_ (95% CI) = 0.87 (0.61–1.24; *p* for trend = 0.443). On the contrary, the association between the LRL score and CRC risk was significantly higher in non-prevalent T2D participants (HR _model 2_ (95% CI) = 0.71 (0.52-0.95; *p* for trend = 0.021) than in individuals with T2D at baseline (HR _model 2_ (95% CI) = 0.81 (0.62–1.06; *P* for trend = 0.443).

## 4. Discussion

This prospective cohort study showed that adherence to the most recent 2018 WCRF/AICR recommendations, as well as accomplishing a cluster of LRL factors, was inversely associated with CRC incidence in elderly Mediterranean individuals at high cardiovascular risk. As far as we are aware, this was the first study to assess the relationship between the recent LRL score [7] and the risk of CRC. On the other hand, this work provided additional support to confirm that new 2018 WCRF/AICR recommendations also apply for aging individuals at high CVD risk.

In agreement with the results obtained, El Kinany and coworkers [16] found that greater adherence to the 2018 WCRF/AICR score decreased overall CRC risk by 42% after comparing 1516 cases with 1516 controls in Morocco. Similarly, a 24-year-follow-up cohort study [13], with 68,977 women from the Nurses’ Health Study and 45,442 men from the Health Professionals Follow-up Study, found a statistically significant inverse association between adherence to the updated WCRF/AICR score and the incidence of CRC risk in men. Furthermore, associations in women were weaker. On the other hand, we did not find any statistically significant differences between sexes in our subgroup analyses for the WCRF/AICR score. However, differences in the sample size, number of CRC cases, and score operationalization between both studies might account for this discrepancy.

Our results confirmed the findings of a meta-analysis of seven prospective cohort studies [26], including 361,616 European and US adults (aged > 60, 43% women) and 6507 CRC cases, within the Consortium on Health and Ageing: Network of Cohorts in Europe and the United States (CHANCES) Project. It showed with a low level of heterogeneity that adherence to WCRF/AICR recommendations for cancer prevention was also applicable in the elderly. Nonetheless, our findings added new evidence to suggest that not only might elderly individuals benefit from following these recommendations but also individuals who are at high CVD risk.

Although some longitudinal studies [6,9,14,15,27] on the former 2007 WCRF/AICR recommendations [28] have also observed statistically significant inverse associations with CRC risk, non-significant results have been reported in other studies [10,11,12,29]. This inconsistency could be due to differences in the study design, different cut-points used, and the number of components for scoring between studies. It should be noted that the 2018 WCRF/AICR report [2] again recommends the intake of at least 30 g/day of fiber and reducing the consumption of processed food high in fat and sugars, but no longer discourages the consumption of energy-dense foods without taking into consideration their nutritional composition [30].

Even though the LRL score was developed in an attempt to assess the impact of an overall healthy lifestyle pattern on all-cause mortality (including cancer and CVD mortality), our results suggested that this score might also be a useful tool for CRC prevention. This is of great potential importance since non-communicable diseases, such as CVD and cancer, are the leading causes of mortality worldwide [31]. 

In our study, even though the reduction in CRC risk due to greater adherence to the 2018 WCRF/AICR score and the LRL index was similar (48% and 52%, respectively), some differences between the two indexes should be examined. In the WCRF/AICR score, there was no penalty for tobacco use, which could have modified the inverse association observed. On the other hand, the penalty for alcohol intake was stricter in the WCRF/AICR score than in the LRL score. The WCRF/AICR advises not to consume alcohol and focuses on dietary national guidelines in the case of intake, whereas the LRL score accepts a moderate intake as adhering fully to the alcohol recommendation. These different cut-off points for alcohol intake might also have had different roles in the associations found. Furthermore, the WCRF/AICR score gives more weight to nutritional factors, while the LRL score considers the overall diet as a single component using the AHEI-2010 score and, therefore, does not ignore the synergy between nutritional components.

Regarding subgroup analyses, considering that no significant interactions were found with CRC risk, and because confidence intervals strongly overlapped between subgroups, we could not assure a significant interaction between the scores and CRC risk based on age, sex, and diabetes status.

In terms of the WCRF/AICR score, greater adherence to the sugar-sweetened beverages recommendation was strongly and inversely associated with a statistically significant decreased risk of developing CRC of 58%. Because neither the other individual recommendations of the WCRF/AICR score nor the individual components of the LRL score were significantly associated with CRC risk, it seemed that a synergistic effect between components might be suggested. However, most of the non-significant associations between the components of both scores and CRC risk were in the expected direction.

Our results suggested that synergy between the various factors of each score might be one of the main mechanisms. On the other hand, sugar-sweetened beverages consumption had been shown to be independently associated with CRC risk in our analyses. It should be noted that according to the WCRF/AICR, sugar-sweetened beverages intake is mostly linked to weight gain, which increases CRC risk, while no specific components in the drinks are mentioned that may increase the risk of this cancer. However, other potential mechanisms suggested in the literature should be further explored.

High glycemic index and glycemic load from sugar-sweetened beverages might stimulate postprandial glucose and/or insulin response, which is associated with diabetes-related cancer risk [32]. The evidence suggests that insulin stimulates tumor growth by decreasing the production of insulin-like growth factor-binding protein 1 (IGFBP-1) [33], increasing the tissue bioavailability of Insuline-like growth factor (IGF)-I, or inhibiting apoptosis [34]. Furthermore, some in vitro studies have shown that insulin might also stimulate the mitogenesis of cultured normal colorectal epithelial cells and tumor angiogenesis [35,36]. In addition, it has been reported that consumers of sugar drinks may be exposed to 4-methylimidazole (4-MEI), a potential carcinogen formed during the manufacture of the caramel color, added to many widely consumed beverages as a colorant. More studies are needed to investigate how this chemical contributes to the risk of CRC [37].

The findings of the present study have to be interpreted in light of some limitations. Firstly, score components were added, so they contributed equally to the total score. The fact that not all the components of the WCRF/AICR score are associated with the risk of developing CRC might have weakened the associations observed. Since CRC was a secondary outcome in the PREDIMED trial, we did not have enough CRC cases to evaluate CRC subtypes separately (colon and rectum) or by tumor site (left and right colon) with sufficient statistical power. This lack of statistical power might also explain the lack of association found between individual components consistently associated with decreased CRC. Because of insufficient data, three items from the WCRF/AICR score (breastfeeding, cancer prevention supplement use, and following the recommendations after a cancer diagnosis) were omitted. Nonetheless, it is important to mention that the breastfeeding component is not mandatory since it is only applicable to a specific subpopulation. In addition, multivitamin supplement use in our cohort was extremely low. Therefore, this component would have had very little impact on our results. Besides, we also decided to omit supplement vitamin use and the cancer survivors component because of operational redundancy [30]. Additionally, a generalization of our results might be limited because of our specific study population at high cardiovascular risk. Further, although the intervention arm was considered as a potential confounder in our main statistical models, we could not rule out residual confounding due to changes in diet between groups. Finally, the causality of the observed associations could not be established due to the observational nature of our study.

However, our study had several strengths. First, it had a large-scale prospective design, a long follow-up period, and a large sample. Second, a Clinical Event Adjudication Committee confirmed all major events annually. Third, we used a validated FFQ and, although residual confounding control could not be ruled out, we controlled for several potential confounders in our statistical analyses. Fourth, we used data-driven approaches (quantiles and median cut-offs) to operationalize recommendations that did not provide quantitative cut-offs, and a three-level WCRF/AICR score, which gives a more continuous variable and considers partial adherence. Finally, we confirmed the robustness of our results by testing our primary results with two different approaches—multivariable time-dependent Cox proportional regression models and mixed-effects Cox regression models—and stratified analyses.

To conclude, following the 2018 WCRF/AICR recommendations and healthier behavior patterns, the calculated LRL score could substantially help to reduce the risk of developing CRC in an elderly Mediterranean population at high CVD risk. Although the WCRF/AICR score was specifically designed in the context of cancer prevention, we suggest that using other scores based on healthy lifestyle patterns, such as the LRL index, might also be a useful tool to prevent CRC, especially in elderly individuals. The results of this study need to be replicated in other populations with different dietary and lifestyle patterns. For this purpose, large-prospective cohorts with more events are needed to extend the evidence on lifestyle patterns and CRC risk.

## Figures and Tables

**Figure 1 jcm-09-01215-f001:**
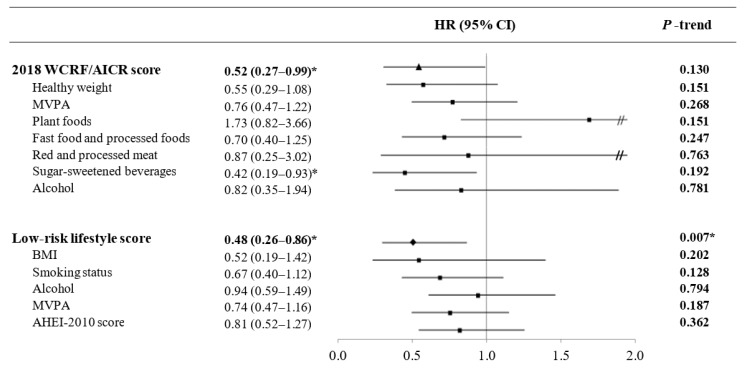
Colorectal cancer risk associated with 2018 WCRF/AICR and low-risk lifestyle scores and the individual components of each index in the PREvención con DIeta MEDiterránea (PREDIMED) cohort (*n* = 7216). Multivariable Cox proportional regression models were used. Results were the HRs (95% CIs) for the comparison between the highest vs. the lowest quantile for each overall score (2018 WCRF/AICR score: Quartile (Q) 4 vs. Q1; low-risk lifestyle score: Tertile (T) 3 vs. T1) and the comparison for the highest vs. the lowest category for each individual component of the score (see Table 3 and Table 4). * *p*-value < 0.05. *p* for trend stands for linear trend. Abbreviations: AHEI, alternate healthy eating index; BMI, body mass index; CI, confidence interval; HR, hazard ratio; MVPA, moderate-to-vigorous physical activity; WCRF/AICR, World Cancer Research Fund/American Institute for Cancer Research. The triangle represents the HR for the WCRF/AICR score; the square represents the HRs for the individual components of each score, and the diamond represents the HR for the LRL score; Double slash appears when CI > 2.

**Table 1 jcm-09-01215-t001:** Baseline characteristics of the study population in the PREvención con DIeta MEDiterránea (PREDIMED) study ^a^.

	Total Study Population *n* = 7216	Colorectal Cancer Events *n* = 97	Non-Events *n* = 7119	*p*-Value ^b^
2018 WCRF/AICR score	3.8 (3.3–4.4)	3.8 (3.2–4.2)	3.8 (3.3–4.4)	0.110
Low-risk lifestyle score	2 (1–2)	1 (1–2)	2 (1–2)	0.021 *
Age, years	67 (62–72)	67 (62–72)	67 (62–72)	0.269
Women, % (*n*)	57.4 (4145)	41.2 (40)	57.7 (4105)	0.001 *
Education level, % (*n*)				
Primary, secondary, high school	92.9 (6700)	92.8 (90)	92.9 (6610)	0.980
University/graduate	7.2 (516)	7.2 (7)	7.2 (509)	
Age at diagnosis of cancer, years	72.4 (66.8–76.4)	72.4 (66.8–76.4)	–	–
Family history of cancer, % (*n*)	49.2 (3548)	41.2 (40)	49.3 (3508)	0.116
Cancer location				
Colon, % (*n*)	77 (79.4)	77 (79.4)	–	–
Rectum, % (*n*)	20 (20.6)	20 (20.6)	–	–
Diabetes, % (*n*)	48.9 (3527)	51.6 (50)	48.8 (3477)	0.597
Hypertension, % (*n*)	82.7 (5970)	80.4 (78)	82.8 (5892)	0.543
Waist circumference, cm				
Women	99 (91–105)	98.2 (92–107)	99 (91–105)	0.614
Men	103 (97–109)	103 (96–11)	103 (97–109)	0.441
BMI, kg/m^2^	29.7 (27.2–32.5)	29.9 (27.6–32.3)	29.7 (27.2–32.5)	0.813
MVPA, min./wk.	42.2 (0–296.5)	52.5 (0–311)	42.1 (0–296.5)	0.982
Smoking status, % (*n*)				
Never smokers	61.5 (4438)	47.4 (46)	61.7 (4392)	0.011 *
Former smokers	24.6 (1774)	30.9 (30)	24.5 (1744)	
Current smokers	13.9 (1004)	21.7 (21)	13.8 (983)	
Current medication, % (*n*)				
Aspirin	22.4 (1613)	24.7 (24)	22.3 (1589)	0.758
HRT (only in women)	2.8 (115)	5 (2)	2.8 (113)	0.234
Intervention groups, % (*n*)				
MedDiet + EVOO	34.3 (2474)	39.2 (38)	34.2 (2436)	0.390
MedDiet + nuts	32.7 (2360)	34.0 (33)	32.7 (2327)	
Control low–fat diet	33.0 (2382)	26.8 (26)	33.1 (2356)	
Energy intake (kcal/day)	2184.4 (1842.9–2579.7)	2183.4 (1942–2669.3)	2184.5 (1841.6–2578.2)	0.263
AHEI–2010 score	64.4 (58.8–70.2)	62.6 (57.4–69.1)	64.4 (58.9–70.2)	0.210
Food consumption, g/day				
Vegetables	313.7 (235–408.7)	289.2 (221.2–367)	314 (235–409.3)	0.164
Fruits	333.1 (227.2–475)	324.5 (222.4–459.2)	333.3 (227.6–475)	0.454
Legumes	17.1 (12.6–25.1)	17.1 (16–25.1)	17.1 (12.6–25.1)	0.778
Red and processed meat	68.1 (42.6–100)	68.3 (45.7–111.7)	68.1 (42.4–100)	0.073
Fast food and processed foods	72.4 (45.7–109.2)	77.2 (51–108.6)	72.4 (45.7–109.2)	0.746
Sugar-sweetened beverages	13.3 (0–85.7)	13.3 (0–99.1)	13.3 (0–85.7)	0.388
Alcohol	1.5 (0–10.4)	4.3 (0–12.2)	1.5 (0–10.4)	0.093

^a^ Data are expressed as medians (IQR, interquartile range) for continuous variables and percentage and number (*n*) for categorical variables. ^b^
*p*-values for comparison between colorectal cancer cases and non-cases were calculated by chi-square or *t*-Student tests for categorical and continuous variables, respectively. All statistical tests were two-sided. * *p*-value <0.05. Abbreviations: AHEI, alternate healthy eating index; BMI, body mass index; EVOO, extra virgin olive oil; HRT, hormone replacement therapy; MedDiet, Mediterranean diet; min/wk., min/week; MVPA, moderate-to-vigorous physical activity; WCRF/AICR, World Cancer Research Fund/American Institute for Cancer Research.

**Table 2 jcm-09-01215-t002:** HRs and 95% CIs between the 2018 WCRF/AICR and the low-risk lifestyle scores and colorectal cancer risk at baseline in the PREDIMED study (*n* = 7216).

	2018 WCRF/AICR Score		Low-Risk Lifestyle Score	
	Continuous Analysis (1-Point Increment), HR (95% CI)	*p* for Trend	Continuous Analysis (1-Point Increment), HR (95% CI)	*p* for Trend
Events/non-events (*n*)	97/7216	-	97/7216	-
Crude model	0.79 (0.63–0.98) *	0.033 *	0.77 (0.62–0.95) *	0.016 *
Model 1	0.78 (0.62–0.98) *	0.034 *	0.78 (0.64–0.96) *	0.019 *
Model 2	0.79 (0.63–0.99) *	0.045 *	0.78 (0.64–0.96) *	0.017 *

Model 1 adjusted for age (years - continuous) and sex. Model 2 was model 1 plus intervention group (MedDiet + EVOO, MedDiet + nuts, low-fat control), family history of cancer (yes/no), education level (primary or secondary/high school, university or graduate), history of diabetes (yes/no), baseline energy intake (Kcal/day, continuous), and treatment with aspirin (yes/no) at baseline. Model 2 for the 2018 WCRF/AICR score was further adjusted for a current smoker (yes/no), former smoker (yes/no), never smoker (yes/no). All models were stratified by recruitment center. * *p*-value < 0.05. Abbreviations: CI, confidence interval; EVOO, extra virgin olive oil; HR, hazard ratio; MedDiet, Mediterranean Diet; WCRF/AICR, World Cancer Research Fund/American Institute for Cancer Research.

**Table 3 jcm-09-01215-t003:** Mutually adjusted HRs and 95% CIs for colorectal cancer risk associated with categories of individual components of the 2018 WCRF/AICR score at baseline in the PREDIMED study (*n* = 7216).

	Component Score, HR (95% CI)				
**Healthy weight**	**0**	**0.25**	**0.5**	**>0.75**	***p* for trend**
Events/non-events (*n*)	22/1381	22/1723	33/2233	20/1782	-
Crude model	1.00	0.77 (0.41–1.45)	0.90 (0.50–1.60)	0.62 (0.33–1.17)	0.220
Model 1	1.00	0.71 (0.37–1.33)	0.85 (0.48–1.54)	0.54 (0.28–1.06)	0.135
Model 2	1.00	0.71 (0.38–1.33)	0.87 (0.48–1.55)	0.54 (0.28–1.05)	0.134
Model 3	1.00	0.71 (0.38–1.35)	0.88 (0.49–1.56)	0.55 (0.29–1.08)	0.151
**Physical activity**	**0**	**0.5**	**1**	**-**	***p* for trend**
Events/non-events (*n*)	53/3893	11/590	33/2636	-	
Crude model	1.00	1.25 (0.63–2.48)	0.82 (0.51–1.32)	-	0.441
Model 1	1.00	1.25 (0.63–2.48)	0.75 (0.46–1.22)	-	0.254
Model 2	1.00	1.24 (0.62–2.48)	0.76 (0.47–1.22)	-	0.271
Model 3	1.00	1.24 (0.63–2.46)	0.76 (0.47–1.22)	-	0.268
**Plant foods**	**0-0.5**	**>0.5–0.75**	**>0.75**	**-**	***p* for trend**
Events/non-events (*n*)	14/1345	56/3952	27/1822	-	
Crude model	1.00	1.38 (0.77–2.47)	1.66 (0.82–3.34)	-	0.158
Model 1	1.00	1.34 (0.75–2.40)	1.57 (0.78–3.16)	-	0.210
Model 2	1.00	1.34 (0.75–2.41)	1.60 (0.77–3.32)	-	0.206
Model 3	1.00	1.38 (0.76–2.49)	1.73 (0.82–3.66)	-	0.151
**Fast food and processed foods**	**0**	**0.5**	**1**	**-**	***p* for trend**
Events/non-events (*n*)	18/1201	39/2766	40/3152	-	-
Crude model	1.00	0.82 (0.46–1.45)	0.72 (0.41–1.26)	-	0.258
Model 1	1.00	0.84 (0.47–1.49)	0.72 (0.41–1.28)	-	0.260
Model 2	1.00	0.85 (0.48–1.49)	0.73 (0.41–1.29)	-	0.278
Model 3	1.00	0.81 (0.47–1.42)	0.70 (0.40–1.25)	-	0.247
**Red and processed meat**	**0**	**0.5**	**1**	**-**	***p* for trend**
Events/non-events (*n*)	36/2263	58/4577	3/279	-	-
Crude model	1.00	0.89 (0.58–1.36)	0.77 (0.23–2.55)	-	0.531
Model 1	1.00	0.95 (0.60–1.48)	0.87 (0.26–2.94)	-	0.771
Model 2	1.00	0.97 (0.62–1.54)	0.90 (0.26–3.07)	-	0.868
Model 3	1.00	0.94 (0.60–1.48)	0.88 (0.25–3.06)	-	0.771
**Sugar-sweetened beverages**	**0**	**0.5**	**1**	**-**	***p* for trend**
Events/non-events (*n*)	8/326	48/3675	41/3118	-	-
Crude model	1.00	0.45 (0.21–0.96) *	0.45 (0.21–0.94) *	-	0.298
Model 1	1.00	0.46 (0.22–0.98) *	0.43 (0.20–0.90) *	-	0.174
Model 2	1.00	0.47 (0.21–1.02)	0.43 (0.20–0.94) *	-	0.185
Model 3	1.00	0.45 (0.21–1.00)	0.42 (0.19–0.93) *	-	0.192
**Alcohol intake**	**0**	**0.5**	**1**	**-**	***p* for trend**
Events/non-events (*n*)	7/389	60/4111	30/2619	-	-
Crude model	1.00	0.86 (0.40–1.86)	0.68 (0.31–1.52)	-	0.221
Model 1	1.00	0.86 (0.40–1.85)	0.85 (0.38–1.89)	-	0.771
Model 2	1.00	0.96 (0.43–2.12)	0.98 (0.42–2.27)	-	0.979
Model 3	1.00	0.82 (0.36–1.87)	0.82 (0.35–1.94)	-	0.781

Model 1 adjusted for age (years - continuous) and sex. Model 2 was further adjusted for the intervention group (MedDiet + EVOO, MedDiet + nuts, low-fat control), current smoker (yes/no), former smoker (yes/no), never smoker (yes/no), family history of cancer (yes/no), education level (primary or secondary/high school, university or graduate), history of diabetes (yes/no), baseline energy intake (Kcal/day, continuous), and treatment with aspirin (yes/no) at baseline. Model 3 was model 2 plus the other individual components at baseline: healthy weight (0, 0.25, 0.5, >0.75 point), physical activity (0, 0.5, 1 point), plant foods (0–0.5, >0.5–0.75, >0.75 point), fast food and processed foods (0, 0.5, 1 point), red and processed meat (0, 0.5, 1 point), sugar-sweetened beverages (0, 0.5, 1 point), and alcohol intake (0, 0.5, 1 point). Categories for each score component were established based on the number of subcomponents and the score distribution in the population studied. All models were stratified by recruitment center. * *p*-value < 0.05. Abbreviations: CI, confidence interval; EVOO, extra virgin olive oil; HR, hazard ratio; MedDiet: Mediterranean diet; WCRF/AICR, World Cancer Research Fund/American Institute for Cancer Research.

**Table 4 jcm-09-01215-t004:** Mutually adjusted HRs and 95% CIs for colorectal cancer risk associated with different categories of individual low-risk lifestyle components at baseline in the PREDIMED study (*n* = 7216).

	Component Score, HR (95% CI)		
**BMI**	**0**	**1**	***p* for trend**
Events/non-events (*n*)	93/6607	4/512	
Crude model	1.00	0.54 (0.20–1.46)	0.222
Model 1	1.00	0.52 (0.19–1.41)	0.200
Model 2	1.00	0.51 (0.19–1.39)	0.190
Model 3	1.00	0.52 (0.19–1.42)	0.202
**Smoking status**	**0**	**1**	***p* for trend**
Events/non-events (*n*)	51/2727	46/4392	
Crude model	1.00	0.55 (0.36–0.83) *	0.004 *
Model 1	1.00	0.66 (0.40–1.11)	0.119
Model 2	1.00	0.66 (0.39–1.10)	0.113
Model 3	1.00	0.67 (0.40–1.12)	0.128
**Alcohol consumption**	**0**	**1**	***p* for trend**
Events/non-events (*n*)	65/4992	32/2127	
Crude model	1.00	1.12 (0.73–1.71)	0.594
Model 1	1.00	0.90 (0.58–1.39)	0.626
Model 2	1.00	0.88 (0.56–1.39)	0.594
Model 3	1.00	0.94 (0.59–1.49)	0.794
**Physical activity**	**0**	**1**	***p* for trend**
Events/non-events (*n*)	64/4483	33/2636	
Crude model	1.00	0.79 (0.51–1.24)	0.303
Model 1	1.00	0.72 (0.45–1.13)	0.155
Model 2	1.00	0.71 (0.45–1.12)	0.146
Model 3	1.00	0.74 (0.47–1.16)	0.187
**AHEI-2010 score**	**0**	**1**	***p* for trend**
Events/non-events (*n*)	64/4266	33/2853	0.268
Crude model	1.00	0.81 (0.52–1.25)	0.269
Model 1	1.00	0.78 (0.51–1.21)	0.268
Model 2	1.00	0.78 (0.50–1.21)	0.268
Model 3	1.00	0.81 (0.52–1.27)	0.362

Model 1 adjusted for age (years - continuous) and sex. Model 2 was further adjusted for the intervention group (MedDiet + EVOO, MedDiet + nuts, low-fat control), family history of cancer (yes/no), education level (primary or secondary/high school, university or graduate), history of diabetes (yes/no), baseline energy intake (Kcal/day, continuous), and treatment with aspirin (yes/no) at baseline. Model 3 was model 2 plus the other individual low-risk lifestyle components at baseline: BMI ≥18.5 and ≤24.9 kg/m^2^ (yes/no), never smoker (yes/no), low-risk alcohol consumption (yes/no), AHEI-2010 score ≥P60 (yes/no), MVPA ≥150 min./wk (yes/no). All models were stratified by recruitment center. * *p*-value <0.05. Abbreviations: AHEI, alternate healthy eating index; BMI, body mass index; CI, confidence interval; EVOO, extra virgin olive oil; HR, hazard ratio; MedDiet: Mediterranean diet; P, percentile.

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
