# Peer review of "Association between the 2018 WCRF/AICR and the Low-Risk Lifestyle Scores with Colorectal Cancer Risk in the Predimed Study"

_jcm, 2020, doi:10.3390/jcm9041215_

Round 1

Reviewer 1 Report

The manuscript by Barrubés et al. reports on the inverse association of the 2018 WCRF/AICR and another low-risk lifestyle score with colorectal cancer risk in an elderly Spanish population at high CVD risk. The finding of a significant association of a healthy lifestyle with reduced colorectal cancer risk in a population that already belongs to a colorectal cancer risk group provide a strong contribution to the literature.

However, I have some major concerns that I would like to raise below:

  • In the abstract and also the manuscript the authors state that this is a longitudinal study. However, from the explanations given for study design the study rather seems to be of prospective design. If this study has a longitudinal component, please describe the study design in more detail to make this point more clear.
  • The last sentence of the introduction states: “Our secondary objective was to evaluate the associated CRC risk for each prevention recommendation.” It is a) not clear what is meant by “each prevention recommendation”, and b) it is not clear how this is addressed in the data analysis.
  • Methods-participant description: The authors state that participants for the PREDIMED trial were included based on high risk for CVD. Which underlying conditions were considered for this inclusion criterion?
  • Methods-statistical analysis: The authors state that they analyzed the scores in quantiles comparing the highest versus the lowest quantile. What was the rationale for this analysis. Also in the results the authors present results for an analysis of the scores as continuous variables and the association with CRC risk with a 1-increment increase in score. This was not mentioned in the methods section.
  • Methods-statistical analysis: On what basis were potential confounders included in the different models?
  • Results: The structure of the results is generally confusing. There is no rationale given for the performed analyses and the reader gets easily confused without a better structure of the paragraphs. One suggestion could be subheadings to enable better understanding of the flow of the results.
  • Results-Table 1: The authors report continuous variables as median with IQR but report a Students t-test for comparison. Were the continuous variables normally distributed? Or would a non-parametric test be more appropriate to compare continuous variables between cases and controls?
  • Results-Figure 1: Figure 1 seems redundant since it presents the same results as presented in table 3 and 4. If kept in the manuscript the figure legend should state the model applied.
  • Results-ll 329-341: The authors here state that they observed significant differences in the association with CRC by subgroup. However, in the methods it is stated that there were no significant interactions observed. Also when looking at the confidence intervals they are strongly overlapping between subgroups. Thus, results for this section are over-interpreted and should be toned down throughout the manuscript.
  • Discussion- ll 400-402: The authors state that sugar-sweetened beverages are the main contributor to the association between WCRF/AICR score and CRC. However, this conclusion cannot be drawn from the performed analysis. In any case, sugar-sweetened beverages would be independently associated with CRC risk from the other variables included in the score. However, this should also be assessed by adjusting the association with the score itself.

Author Response

Response to Reviewer 1 Comments

The manuscript by Barrubés et al. reports on the inverse association of the 2018 WCRF/AICR and another low-risk lifestyle score with colorectal cancer risk in an elderly Spanish population at high CVD risk. The finding of a significant association of a healthy lifestyle with reduced colorectal cancer risk in a population that already belongs to a colorectal cancer risk group provide a strong contribution to the literature.

We sincerely thank the Reviewer 1 for all the valuable suggestions, which have importantly improved the initial version of the manuscript. We have addressed all these suggestions and comments in each of the following points, as well as in the manuscript, when required. Please find below the itemized responses to all Reviewer’s comments.

However, I have some major concerns that I would like to raise below:

Point 1: In the abstract and also the manuscript the authors state that this is a longitudinal study. However, from the explanations given for study design the study rather seems to be of prospective design. If this study has a longitudinal component, please describe the study design in more detail to make this point more clear.

Response 1: Thank you for the suggestion. We described more precisely the study design (line 101)

Point 2: The last sentence of the introduction states: “Our secondary objective was to evaluate the associated CRC risk for each prevention recommendation.” It is a) not clear what is meant by “each prevention recommendation”, and b) it is not clear how this is addressed in the data analysis.

Response 2: We totally agree with these observations. Thus, we clarified this point as the Reviewer can check below.

  1. We better detailed what “each prevention recommendation” is (lines 93-94).
  2. The highest vs the lowest categories for each individual component of each score and their association with CRC risk were compared by using Multivariable time-dependent Cox proportional regression models (detailed in lines 212-223). Results are the hazard ratios (HRs) and their 95% confidence intervals (CIs) for the comparison between the highest vs the lowest quantile for each individual component composing each score. Results of these analyses are presented in Tables 3 and 4.

Point 3: Methods-participant description: The authors state that participants for the PREDIMED trial were included based on high risk for CVD. Which underlying conditions were considered for this inclusion criterion?

Response 3: We really appreciate the reviewer suggestion. In the new m/s, we detailed the underlying conditions considered in order to include PREDIMED participants who were at high CVD (lines 116-119).

Point 4: Methods-statistical analysis: The authors state that they analyzed the scores in quantiles comparing the highest versus the lowest quantile. What was the rationale for this analysis. Also in the results the authors present results for an analysis of the scores as continuous variables and the association with CRC risk with a 1-increment increase in score. This was not mentioned in the methods section.

Response 4: Quantiles for each index were calculated considering the distribution of the variable in the analysed population in order to obtain similar number of participants in each quantile of the score. We aimed to compare those individuals with highest adherences in the scores with those with the lowest ones. Thus, we considered that the most appropriate way to proceed was to compare the highest vs the lowest quantile.

As suggested by the Reviewer, in the new m/s we mentioned how we assessed the continuous analyses in the methods section (lines 221-223).

Point 5: Methods-statistical analysis: On what basis were potential confounders included in the different models?

Response 5: The rationale for the inclusion of each potential confounder in the Cox regression models is detailed below:

- Model 1 was adjusted for demographic variables linked to CRC risk (age and sex).

- Model 2 comprised model 1 additionally adjusted for other sociodemographic and medical conditions that have been shown to be related with the development of CRC risk (for example, educational level, family history of cancer, history of diabetes and medication use). This model was also adjusted for other factors associated with the intervention and diet of each participant that might modify the associations such as the intervention group and the baseline energy intake.

- Model 3 was Model 2 plus the other individual components for each score.

Point 6: Results: The structure of the results is generally confusing. There is no rationale given for the performed analyses and the reader gets easily confused without a better structure of the paragraphs. One suggestion could be subheadings to enable better understanding of the flow of the results.

Response 6: We really appreciate the Reviewer suggestion. Therefore, we added subheadings to facilitate understanding of the results.

Point 7: Results-Table 1: The authors report continuous variables as median with IQR but report a Students t-test for comparison. Were the continuous variables normally distributed? Or would a non-parametric test be more appropriate to compare continuous variables between cases and controls?

Response 7: The continuous variables were normally distributed. To better clarify this point, we added this information in the new m/s (line 196).

Point 8: Results-Figure 1: Figure 1 seems redundant since it presents the same results as presented in table 3 and 4. If kept in the manuscript the figure legend should state the model applied.

Response 8: We appreciate the Reviewer comment. As suggested, we kept the figure in the manuscript and stated the model used for the analyses.

Point 9: Results-ll 329-341: The authors here state that they observed significant differences in the association with CRC by subgroup. However, in the methods it is stated that there were no significant interactions observed. Also when looking at the confidence intervals they are strongly overlapping between subgroups. Thus, results for this section are over-interpreted and should be toned down throughout the manuscript.

Response 9: We thank the Reviewer for this useful comment. As suggested, we toned down the conclusions obtained from the subgroup analyses in the m/s (lines 416-419).

Point 10: Discussion-ll 400-402: The authors state that sugar-sweetened beverages are the main contributor to the association between WCRF/AICR score and CRC. However, this conclusion cannot be drawn from the performed analysis. In any case, sugar-sweetened beverages would be independently associated with CRC risk from the other variables included in the score. However, this should also be assessed by adjusting the association with the score itself.

Response 10: We totally agree with the Reviewer’s comment. Therefore, we reconsidered the conclusion regarding sugar-sweetened beverages (lines 427-431).

Reviewer 2 Report

The authors conducted an observational analysis in the dataset of the PREDIMED randomized controlled trial. They observed that adherence to two different lifestyle scores was associated with risk of CRC cancer. 

Although these findings are not necessarily novel, I believe it is important to show that these associations hold in populations at high risk of CVD.

The paper is written well and is extensive. What I missed in the manuscript is a description of potential changes in diet/lifestyle over time. Especially since this was a trial focussing on changing dietary intake, some of the groups will have changed their intake over time. How will this have affected the adherence to the scores over time? And will this have affected the findings of this study? 

Other comments:

  • in figure 1, for the WCRF score, the HR for the lowest versus the highest are presented. The quartiles are not presented in the as a baseline table. Given that the number of cases is relatively low, it would help to show a baseline table of WCRF score quartiles to see how the cases are distributed, and other baseline characteristics too.
  • same comment for the lifestyle score in figure 1. A baseline table of the lifestyle score in tertiles would be very helpful.
  • Why have the authors decided to present the WCRF score in quartiles, and the lifestyle score in tertile?  
  • in the discussion, the authors discuss why sugar-sweetened beverages could potentially increase the risk of CRC. In the WCRF reports, sugar-sweetened beverages are mostly linked to weight-gain, which increases the risk of CRC, while no specific components in the drinks are mentioned that may increase the risk. Would be good to add that to the paper too.  

Author Response

Response to Reviewer 2 Comments

The authors conducted an observational analysis in the dataset of the PREDIMED randomized controlled trial. They observed that adherence to two different lifestyle scores was associated with risk of CRC cancer. 

Although these findings are not necessarily novel, I believe it is important to show that these associations hold in populations at high risk of CVD.

Point 1: The paper is written well and is extensive.

We sincerely thank the Reviewer 2 for all the valuable suggestions, which have importantly improved the initial version of the manuscript. We have addressed all these suggestions and comments in each of the following points, as well as in the manuscript, when required. Please find below the itemized responses to all Reviewer’s comments.

What I missed in the manuscript is a description of potential changes in diet/lifestyle over time. Especially since this was a trial focussing on changing dietary intake, some of the groups will have changed their intake over time. How will this have affected the adherence to the scores over time? And will this have affected the findings of this study? 

Response 1: We really appreciate the Reviewer suggestion. We acknowledge that potential changes in diet might have occurred due to the nutritional intervention because nutritional recommendations were made in the intervention groups. Therefore, we address this fact in the discussion section: “although the intervention arm was considered as a potential confounder in our main statistical models, we cannot rule out residual confounding due to changes in diet between groups” (lines 456-458).

On the other hand, it should be noted that no recommendations were given regarding physical activity or other lifestyle factors during the intervention. Thus, we do not think that potential changes in lifestyle (except for diet) might have occurred.

Other comments:

Point 2: in figure 1, for the WCRF score, the HR for the lowest versus the highest are presented. The quartiles are not presented in the as a baseline table. Given that the number of cases is relatively low, it would help to show a baseline table of WCRF score quartiles to see how the cases are distributed, and other baseline characteristics too.

Point 3: same comment for the lifestyle score in figure 1. A baseline table of the lifestyle score in tertiles would be very helpful.

Responses 2 and 3: We really appreciate this interesting suggestion, and we consider that this will improve our m/s quality. Therefore, we added Table S3 and Table S4 (supplementary material) and stated this in lines 294-296.

Point 4: Why have the authors decided to present the WCRF score in quartiles, and the lifestyle score in tertile?  

Response 4: Quantiles for each index were calculated considering the distribution of the variable in the analyzed population in order to obtain similar number of participants in each quantile of the scores (lines 201-202).

Point 5: in the discussion, the authors discuss why sugar-sweetened beverages could potentially increase the risk of CRC. In the WCRF reports, sugar-sweetened beverages are mostly linked to weight-gain, which increases the risk of CRC, while no specific components in the drinks are mentioned that may increase the risk. Would be good to add that to the paper too.  

Response 5: We totally agree with the Reviewer suggestion. Thus, we added this conclusion in the discussion section (lines 426-431).

Round 2

Reviewer 1 Report

The authors have adequately adressed my suggestions. I have no further comments.

Author Response

Dear Reviewer 1,

Thank you very much for all the suggestions on our manuscript. We really think that all the major revisions really improved the quality of our work.

Yours sincerely.

Dr. Nancy Babio

Dr. Pablo Hernández-Alonso